# Optimal Consensus with Dual Abnormality Mode of Cellular IoT Based on Edge Computing

**DOI:** 10.3390/s21020671

**Published:** 2021-01-19

**Authors:** Shin-Hung Pan, Shu-Ching Wang

**Affiliations:** 1Department of M-Commerce and Multimedia Applications, Asia University, Taichung 413305, Taiwan; vincentpan@asia.edu.tw; 2Department of Information Management, Chaoyang University of Technology, Taichung 413310, Taiwan

**Keywords:** Internet of Things, cellular internet of things, edge computing, cloud computing, fault-tolerant, consensus problem

## Abstract

The continuous development of fifth-generation (5G) networks is the main driving force for the growth of Internet of Things (IoT) applications. It is expected that the 5G network will greatly expand the applications of the IoT, thereby promoting the operation of cellular networks, the security and network challenges of the IoT, and pushing the future of the Internet to the edge. Because the IoT can make anything in anyplace be connected together at any time, it can provide ubiquitous services. With the establishment and use of 5G wireless networks, the cellular IoT (CIoT) will be developed and applied. In order to provide more reliable CIoT applications, a reliable network topology is very important. Reaching a consensus is one of the most important issues in providing a highly reliable CIoT design. Therefore, it is necessary to reach a consensus so that even if some components in the system is abnormal, the application in the system can still execute correctly in CIoT. In this study, a protocol of consensus is discussed in CIoT with dual abnormality mode that combines dormant abnormality and malicious abnormality. The protocol proposed in this research not only allows all normal components in CIoT to reach a consensus with the minimum times of data exchange, but also allows the maximum number of dormant and malicious abnormal components in CIoT. In the meantime, the protocol can make all normal components in CIoT satisfy the constraints of reaching consensus: Termination, Agreement, and Integrity.

## 1. Introduction

In the past few years, many 5G technologies have been developed to provide new infrastructure and design as well as the functions required by the future Internet of Things (IoT) [1,2]. The 5G-based IoT can provide real-time, on-demand, online, and reconfigurable use for applications. The IoT has been widely used in various fields, such as: Smart city, smart environment, smart water, smart metering, industrial control, smart agriculture, smart animal breeding, and smart health [3,4]. In addition, because 5G cellular technology has high-speed transmission capabilities, it will be very suitable to apply to the IoT environment. Therefore, cellular technology has a very important impact on the development of related applications of the IoT [5,6].

Vukobratovic et al. proposed a feasible cellular IoT (CIoT) topology in 2019, which can integrate edge computing and IoT into cellular networks to provide a highly flexible CIoT platform [7]. The CIoT platform proposed by Vukobratovic et al. will be applied and redefined as ECIoT (edge-computing-based CIoT) in this research. Because the advantages of edge computing have been applied to ECIoT, the high-QoS (Quality of Service) IoT applications will be provided [8]. In other words, with the feature of localized computing functions provided by edge computing, ECIoT can provide higher-performance services for IoT-related applications.

In a distributed computing environment CIoT [9], processing elements (PEs) can provide more resources and computing power through connection and cooperation with each other, so that the efficiency or reliability of computing can be improved. However, in many cases, there may be many abnormal PEs in the distributed computing environment, which makes the distributed computing environment unable to provide a highly reliable services. In addition, many related application services require such a consensus, so reaching a consensus in a distributed computing environment with abnormal PEs is one of the core issues in providing high-reliability computing [9]. For users of the applications of CIoT, the system must provide better performance and reliability [5]. Therefore, in order to ensure the reliability of CIoT-related applications, a protocol must be proposed to allow a set of normal PEs to reach a consensus value [10,11].

With the 5G network, the application of the IoT will greatly expand to promoting the operation of cellular networks and pushing the future of the Internet to the edge [1]. Federated learning (FL) is a model of machine learning in distributed systems. In the research of Savazzi et al. [12], the proposed FL algorithms leverage the cooperation of devices that perform data operations inside the network by iterating local computations and mutual interactions via consensus-based methods. The approach lays the groundwork for integration of FL within 5G and beyond networks characterized by decentralized connectivity and computing, with intelligence distributed over the edge devices. In the study of Lin et al. [13], a practical collaboration infrastructure for 5G network slice broker is designed, where the core challenge is the consensus protocol to guarantee the security and performance of the overall system. By solving the consensus problem, many related applications can be realized, such as the adaptive weighted replication [14,15], information retrieval [16,17], and the flight control system [18,19]. In addition, the consensus problem has also been studied and widely used in various fields such as blockchain and IoT [20,21].

In ECIoT, there are many interconnected PEs. Even if some PEs are abnormal, the normal PEs need to reach a consensus to make the system still work correctly. In this study, the consensus problem of dormant abnormal PEs and malicious abnormal PEs in ECIoT is reconsidered. The main contribution of this research is to solve the consensus problem of PEs in dual abnormality mode, in which both dormant abnormal PEs and malicious abnormal PEs are existed simultaneously in the system. The protocol proposed in this research, Optimal Consensus with Dual Abnormality Mode (OCDAM), can make all normal PEs satisfy the constraints of reaching consensus: Termination, Agreement, and Integrity. Besides, the protocol can make all normal components in ECIoT reach a consensus with the minimum times of data exchange and tolerate the maximum number of abnormal PEs. In other words, the reliability of the system will be maximized.

This study is divided into seven parts. In Section 1, the motivation and goals of this research are given. In Section 2, the background of consensus problem and the comparisons of consensus protocols in different network topologies will be reviewed and compared. The topology of ECIoT is defined in Section 3. The detailed description of the proposed OCDAM is explained in Section 4. In Section 5, an example to illustrate the operation of the proposed OCDAM is given. The correctness and complexity of the proposed protocol will be demonstrated in Section 6. Section 7 is the conclusion and future works of this research

## 2. Related Works

In this section, the background of consensus problem and the comparisons of previous consensus protocols in different network topologies will be discussed explicitly.

### 2.1. The Background of Consensus Problem

The definition of a consensus problem is that when some PEs may be abnormal in a distributed environment, all normal PEs must reach a consensus. That is, the goal of consensus is to obtain a consensus value for normal PEs. The consensus problem is defined as: Each PE chooses an initial value as a starting point and communicates with other PEs by exchanging data. Based on most of the previous research [10,11,22,23,24,25,26,27] and books [28,29,30], the solution to the consensus problem is defined as a protocol that meets the following constraints:

**Termination**: All normal PEs eventually decide on some value.

**Agreement**: Every normal PE decides on the same value.

**Integrity:** If the initial value of each normal *PE_i_* is *v_i_*, all normal PEs should agree on the value *v_i_*.

In ECIoT, it is composed of many PEs, and some PEs may not always operate normally. If the PE can follow the protocol specification during the execution of the consensus protocol, it means that the PE is normal. Otherwise, the PE is considered to be abnormal. There are two symptoms of PE abnormality, namely, dormant abnormality and malicious abnormality [24]. The dormant abnormalities of PE include crashes and omissions. When the PE is permanently disconnected, it can be said that the PE has a crash abnormality. When PE is temporarily unable to send or receive data on time or at all, an omission abnormality will occur. However, if the protocol uses Manchester code [31] to properly encode the exchanged data before transmission, the receiver PE can always identify the dormant abnormality. The behavior of malicious abnormal PE is unpredictable and incredible.

However, in the ECIoT, the characteristics of the connected topology are very important. Therefore, to solve the consensus problem on the ECIoT, the following assumptions are made in this research:Each PE in ECIoT can be uniquely identified.According to the research of Fisher and Lynch [10], in a distributed computing system with *n* PEs (*n* ≥ 4), at most one-third of the PEs can be abnormal, but the system will not be interrupted.The sending PE of the data can always be identified by the receiving PE.

According to the assumptions of this research, the proposed protocol OCDAM can use the minimum times of data exchange and can tolerate the maximum number of dormant and malicious abnormal PEs, so that all normal PEs can still reach consensus underlying ECIoT.

### 2.2. The Comparisons of Consensus Protocols in Different Network Topologies

Because the solution of consensus problem is one of the most commonly used methods in the field of providing reliable distributed systems, many protocols have previously been proposed to solve the consensus problem for different application areas, such as multi-agent systems, peer-to-peer networks [32,33]. In this study, we focus on the basic protocol of reaching consensus underlying different network topologies. In previous results of this area, the consensus problem was solved in many network models with various fallible component assumptions, such as a BroadCasting Network (BCN) [23], a Fully Connected Network (FCN) [10,11,22], a Generalize Connected Network (GCN) [24], a MultiCasting Network (MCN) [25], a Cloud Computing environment (CC) [34], an Integrated Fog IoT (IFIoT) [26], and an Edge-computing-based CIoT (ECIoT) [27].

In [23], the network topology is BCN, and the fallible component assumption involves malicious abnormal PEs only. In [10,22], the network topology is FCN, and the fallible component assumption involves only malicious abnormal PEs. In [11], the network topology is FCN, and the fallible component assumption involves dormant and malicious abnormal PEs. In [24], all PEs of the GCN network are grouping with the same number of PEs, groups are fully connected with each other, and the fallible components are focused on malicious abnormal PEs only. In the MCN [25], all PEs are grouping with different number of PEs, the network topology may not be fully connected, and the symptoms of the fallible components are not restricted to malicious abnormal. In [34], the framework is a cloud computing environment, and the fallible components are malicious abnormal PEs only. In [26], the consensus of an IoT platform that integrated fog and cloud computing (IFIoT) was discussed, and the fallible components are dormant and malicious abnormal PEs. Additionally, in [27], the topology is an edge-computing-based CIoT (ECIoT), and the fallible components are malicious abnormal PEs only. Many graceful consensus protocols have been proposed according to the different network model assumptions. In this research, the topology is an ECIoT, and the fallible component assumption of the proposed protocol OCDAM involves dormant and malicious abnormal PEs. Consensus protocols have been proposed to ensure the reliability and fault-tolerance capability. Table 1 shows a comparison of various protocols over different network models, in which *da* represents a dormant abnormality and *ma* represents a malicious abnormality.

As in previous related studies, many results in the consensus problem are restricted to the assumption of malicious abnormal PEs [27] allowed. Based on this restriction, the fault tolerance capability of distributed systems will be unreasonably reduced. In this study, by allowing dormant and malicious abnormal PEs to simultaneously exist in ECIoT, the consensus problem is reviewed to enlarge the fault tolerant capability. The fault tolerance capability of our proposed protocol (Optimal Consensus with Dual Abnormality Mode, OCDAM) is much better than that of Pan and Wang [27] whose protocol (CIoT Agreement Protocol, CIoTAP) can only tolerate malicious abnormal PEs in ECIoT. Table 2 compares the two protocols, where *d* is the number of dormant abnormal PEs, *m* is the number of malicious abnormal PEs, and *n* is the number of PEs in the ECIoT. It can be seen from Table 2 that if dormant anomalies and malicious anomalies can be treated separately, the capability of fault tolerance will be enhanced.

## 3. The Network Structure

In the related applications of CIoT, millions of CIoT PEs can be connected to a single base station (BS) for data collection [35]. In some time-sensitive applications, transmitting the data sensed by CIoT PE directly through the Internet may not meet the time requirement. Therefore, some calculations and data will reside on the cellular BS in the form of edge computing devices (Edge PE) [7]. As a large number of different CIoT application services must be provided, the demand for a variety of CIoT PEs will increase exponentially. In order to meet the requirements of various CIoT application services, it is very important that a reliable and durable connection communication should be provided. In addition, the cellular networks can provide ubiquitous connectivity; it can reduce the possibility of interruptions that may occur in traditional wireless networks.

In recent years, CIoT with edge computing is one of the popular technologies in a cellular system. With the increase in deployment density of CIoT PEs and the diversification of related application services, high-reliability services in cellular networks have become increasingly challenging. In this research, the high reliability of using edge computing to deploy CIoT will be ensured. That is, the highly reliable CIoT platform ECIoT will be discussed in this study. The structure of ECIoT used in this study is shown in Figure 1.

In this study, ECIoT consists of three layers: Access-layer, Edge-layer, and Cloud-layer. The Access-layer is composed of many CIoT PEs. For specific CIoT applications, CIoT PE is used to sense and report the required sensing signal. Figure 2 shows the Access-layer deployed using CIoT PEs. The CIoT PEs within the communication range of a specific BS will connect to the specific BS and send the sensed data to the Edge-layer. Then, the data needed to provide a specific application can be obtained by the Edge PEs in the Edge cloud.

The Edge-layer is formed by a set of Edge clouds, among which the Edge cloud is composed of many Edge PEs. The data required for a specific application are processed by Edge cloud. Some Cloud PEs form a Cloud-layer, and the services related to cloud users are provided by these Cloud PEs. In ECIoT, various types of request data in real life will be collected by a large number of CIoT PEs. By using these huge request data, a wide range of CIoT application services can be implemented.

In this research, ECIoT was established based on edge computing. In order to reduce the workload of the Cloud-layer, and to shorten the response time and save bandwidth, the resources of computing and data storage are bridged closer to the required location. Therefore, in Edge-layer, the required data of a specific application will be analyzed and processed. In ECIoT, the computing and storage resources are provided by edge computing, ECIoT can provide sufficient computing and data storage resources for connected PEs. Therefore, ECIoT is a platform suitable for serving various CIoT applications.

## 4. The Optimal Consensus with Dual Abnormality Mode (OCDAM)

In order to solve the problem that ECIoT may not reach a consensus due to dormant and malicious abnormal PEs existed, OCDAM is proposed by this research. Because the data received from normal PEs should be the same, based on the same data received, each PE can easily achieve the same consensus value. Therefore, through the execution of OCDAM, the interference of data transmitted from abnormal PEs to all normal PEs can be eliminated. When PE performs data exchanges, all data will be encoded using Manchester encoding [31], which can eliminate the influence of dormant abnormal PEs. Then, the influence of malicious abnormal PEs can be eliminated by using special data structures and voting functions. The detailed description of Manchester encoding is provided in Appendix A.

Basically, the principle of OCDAM is to exchange data with each other PEs firstly, then to remove the influence of abnormal PEs by taking the majority of data received from other PEs. As if the lower bound of the number of data exchange is completed, all the influences of abnormal PEs are proven to be removed and then the consensus can be reached. For more details, CIoT PE is used to sense and transmit required sensing signals to the related application services underlying the ECIoT. The sensing data are sent to the corresponding Edge cloud in the Edge-layer by CIoT PEs. Edge PE located in the Edge cloud receives the sensing data sent from CIoT PEs, and then the majority value of the received sensing data is obtained. The majority value of the received data is used as the initial value (*v_i_*) of the Edge *PE_i_*, which will be used to execute OCDAM. When the consensus value of each Edge cloud is obtained, the value is expressed as the result of a specific service. Finally, the consensus value is transmitted to the Cloud-layer by Edge PEs. In ECIoT, Cloud PE is responsible for collecting the results of different specific services in the Cloud-layer, and then the consensus values can be composed to provide an integrated service center for various CIoT applications.

The characteristics of the connection topology will affect the resolution of consensus problem. Therefore, in the past, all protocols on consensus problem were based on the following assumptions [10,11,22,23,24,25,26,27,28,29,30]:(1)The network discussed in the study is synchronous.(2)All PEs of ECIoT (including CIoT PE, Edge PE, and Cloud PE) can be uniquely identified.(3)All transmitted data will be encoded using Manchester code [31] when PE performs data exchange. Therefore, the dormant abnormal PE can be detected.(4)The abnormal state of any PE cannot be known by other PEs.

By assumptions, it can be known that if the PE cannot be identified and is not unique, the receiving PE cannot identify the sending PE for data exchange. It may not be possible to eliminate the influence of abnormal PE, and thus the consensus cannot be reached. Therefore, the assumptions must be satisfied, which is the limitation of the most research on consensus problem.

Firstly, the times of data exchange required to execute OCDAM will be determined. When the required times of data exchange is determined, the consensus protocol OCDAM must perform two stages: Data Gathering Stage and Consensus Decision Stage. The task of the Data Gathering Stage is to collect data from other PEs through ECIoT. In addition, all data are encoded using Manchester code [31]; when PE is transmitting data, the influence of dormant abnormal PEs can be eliminated in the Data Gathering Stage. Then, the data received in the Data Gathering Stage are used by each normal PE to determine the common consensus value in the Consensus Decision Stage.

Underlying the ECIoT environment, malicious and dormant abnormal PEs may simultaneously exist. In order for all normal PEs to reach a consensus value, the influences of malicious and dormant abnormal PEs must be eliminated. As the data exchange is executed by the PE, all exchanged data have been encoded using Manchester code [31]. Therefore, PEs with dormant abnormal can be detected during the Data Gathering Stage. Then, the influence of malicious abnormal PEs can be eliminated in the Consensus Decision Stage. Therefore, the basic strategy of the proposed method to solve the consensus problem is to remove the influence of the dormant abnormal PEs first, and then remove the influence of the malicious abnormal PEs.

In the research of Fischer and Lynch [10] and Wang et al. [26], ⌊(*n* − 1)/3⌋ + 1 has been proved to be the necessary and sufficient times of data exchange to solve the consensus problem, where *n* is the number of PEs in the basic network. Therefore, when OCDAM is executed by the Edge PE, the required times of data exchange σ is ⌊(*n_Ej_* − 1)/3⌋ + 1, where *n_Ej_* is the number of Edge PEs in the Edge cloud *E_j_* at the Edge-layer and *n_Ej_* > 3. Moreover, when OCDAM is executed by Cloud PE, the required times of data exchange σ is ⌊(*n_C_* − 1)/3⌋ + 1, where *n_C_* is the number of PEs in Cloud-layer and *n_C_* > 3. In other words, if the abnormal components include dormant abnormal and malicious abnormal PEs, OCDAM can make all normal PEs in ECIoT reach a consensus; at the same time, it requires the minimum times of data exchange and can tolerate the maximum number of abnormal components. The OCDAM is explained in the following.

The OCDAM proposed in this research includes two stages, in which the influence of dormant abnormal PEs in CIoT will be eliminated in the Data Gathering Stage, and the influence of malicious abnormal PEs in CIoT will be eliminated in the Consensus Decision Stage. The elimination processes of the influence of dormant and malicious abnormal PE in CIoT are shown in Figure 3 and discussed as follows.

### 4.1. Removing the Influence of Dormant Abnormal PEs

During the Data Gathering Stage, the PE can identify the dormant abnormal PE after receiving the data when the protocol uses the Manchester code [31] to properly encode the transmitted data. Therefore, if the PE receives a data transmitted by dormant abnormal PEs, “λ” is used by OCDAM to replace the data received from the dormant abnormal PE.

### 4.2. Mitigating the Influence of Malicious Abnormal PEs

Each PE need to collect enough data from other PEs to make a decision to obtain the consensus value. These received data can be used to reduce the influence of malicious abnormal PE. In the Data Gathering Stage, a hierarchical structure called a data gathering graph (*dg-graph*) is used during data exchanges. The *dg-graph* is a hierarchical structure that is used to store the received data, which is similar to the data structure proposed by Pan and Wang [27]. The *dg-graph* is constructed by a set of nodes. The exchanged data are stored in the node, and the node is marked with the name of the data sending PE. When OCDAM is executed by Edge PEs and Cloud PEs in ECIoT, each normal PE will maintain a *dg-graph*. This research assumes that each PE can correctly identify the PE sending the data. Therefore, when Edge PEs and Cloud PEs execute the Data Gathering Stage of OCDAM, the *dg-graph* will be established based on the information of the data senders. The detailed description of *dg-graph* is provided in Appendix B.

Since all data are encoded with Manchester code before data are exchanged, OCDAM can eliminate the influence of dormant abnormal PEs. In the first time of Data Gathering Stage, each *PE_i_* multicasts its initial value *v_i_*. When a normal PE receives the data, it stores the received value, denoted as *nd*(*i*), in the level 1 of its *dg-graph*. Then each PE broadcasts the data in the first level of its *dg-graph* to other PEs. However, the received data may still be influenced by malicious abnormal PEs. Therefore, OCDAM requires ⌊(*n* − 1)/3⌋ + 1 data exchanges, where *n* is the total number of PEs in the basic network.

After finishing ⌊(*n* − 1)/3⌋ + 1 times of data exchange in the Data Gathering Stage, each PE will execute the Consensus Decision Stage. Subsequently, function VOTE(α) is used to remove the influence of malicious abnormal PEs and a common value is obtained. Since VOTE(α) is a common value, each normal PE can mitigate the influence of malicious abnormal PEs and agree on the value, thus reaching consensus. The detailed definition of the OCDAM is shown in Figure 4.

The purpose of the consensus protocol is to enable each normal PE in the network to reach a consensus. Therefore, in order to reach a consensus, each PE should exchange data with all other PEs. Then, each normal PE collects enough data to determine the consensus value, and the consensus value of each normal PE must be the same. Since the ECIoT discussed in this study is a synchronous network, there is no need to consider the delay of PE in our discussion [10,11,22,23,24,25,26,27]. Therefore, when the PE executes the proposed protocol OCDAM, the PE can receive data from other PEs within a predictable time. If the PE does not receive the data on time, the data must be affected by the abnormal PE.

In this research, the proposed method is used to solve the consensus problem that dormant and malicious abnormalities may occur in PEs of ECIoT. Since ECIoT is a three-layer topology, the proposed method will be processed in a three-layer hierarchical structure, followed by Access-layer, Edge-layer, and Cloud-layer. According to the three-layer architecture of ECIoT, the execution steps of the proposed method are shown in Figure 5.

The method proposed in this research will be activated by CIoT PEs at the Access-layer, and through the CIoT, PEs can obtain the perception data required by specific application services. To execute OCDAM, three parameters are required, σ, *v_i_*, and *n* where σ is the times required to perform the Data Gathering Stage, *v_i_* is the initial value of *PE_i_*, and *n* is the number of PEs participating in the consensus. In order for all normal PEs to reach a consensus, each PE must collect enough exchange data from all other PEs. Through data exchange, normal PEs can collect enough exchange data for the subsequent Consensus Decision Stage.

## 5. The Example of the Proposed Method

Before the protocol being proven, an example of ECIoT is taken to simulate the full steps of the protocol. This simple experiment can show the protocol can make all normal PEs decide on a common value eventually. Besides, every common value decided is one-to-one corresponding to the initial value of each normal PE. The three constraints of reaching consensus had been satisfied.

Taking the system established by ECIoT as an example to execute the proposed method is presented in this section. Figure 6 is an example environment constructed by ECIoT. In this example, there are six CIoT PEs in the communication range of a specific *BS*_1_ at the Access-layer. One is a dormant abnormal PE, one is a malicious abnormal PE, and four are normal PEs. In Edge cloud *E*_1_ of Edge-layer, there are six Edge PEs. Edge PE *e*_11_ is assumed in dormant abnormal and *e*_14_ is assumed in malicious abnormal. Cloud PE *c*_5_ is a dormant abnormal PE and *c*_4_ is a malicious abnormal PE in Cloud-layer. Furthermore, there are six Cloud PEs in Cloud-layer.

In the proposed method, the Manchester code [31] is used to encode the transmitted data, so the data routed through the dormant abnormal PE can be detected. Therefore, the data sent by the dormant abnormal PE can be detected, and the received data are replaced with λ. At the same time, the behavior of malicious abnormal PE is unpredictable, arbitrary, and undetectable.

Follow the steps shown in Figure 5. First, each CIoT PE in the Access-layer senses the monitoring status. For example, there are six CIoT PEs within the communication range of a specific *BS*_1_, and these six CIoT PEs sense 1, 0, 1, 1, 1, and λ, respectively. Figure 7 is an example of the communication range of a specific *BS*_1_. Then, the sensing monitoring data are transferred from CIoT PEs to the Edge PE in the Edge cloud *E*_1_ of Edge-layer.

The sensing data sent by the CIoT PEs within the communication range of the specific *BS*_1_ are received by the Edge PE in the Edge cloud *E*_1_. If Edge PE receives the sensing data sent by six CIoT PEs as (1,0,1,1,1,λ), these data will be calculated by Edge PE with a majority function (majority(1,0,1,1,1,λ) = 1). Then, the number of times required to perform the Data Gathering Stage in OCDAM (σ = ⌊(*n_Mj_* − 1)/3⌋ + 1 = ⌊(6 − 1)/3⌋ + 1 = 2) is calculated. Next, the OCDAM is executed, the majority value (1) is used as the initial value (*v_i_*) of PE in the Edge cloud *E*_1_, and OCDAM(σ, *v_i_*, *n_Mj_*) = OCDAM (2, 1, 6) is executed. The initial value of each Edge PE in the Edge cloud *E*_1_ at the Edge-layer is shown in Figure 8a.

Then, OCDAM is executed by each Edge PE in the Edge cloud *E*_1_. During the first time of data exchange in the Data Gathering Stage, each Edge PE in the Edge cloud *E*_1_ sends the initial value to all other Edge PEs of the Edge cloud *E*_1_ and receives *n_M_*_1_ (=6) data from other Edge PEs. The data are stored in level 1 of the corresponding *dg-graph* of each Edge PE, as shown in Figure 8b. During the second data exchange, each Edge PE sends the data of level 1 in its *dg-graph* to other Edge PEs in the Edge cloud *E*_1_ and stores the received data at the level 2 of its *dg-graph* in the *n_M_*_1_ (=6) nodes. Figure 8c,d shows the *dg-graphs* established by Edge PE *e*_12_ and *e*_13_ during the Data Gathering Stage, respectively.

Subsequently, in the Consensus Decision Stage, the function VOTE(α) is applied to the level 1 of the *dg-graph* with each Edge PE to obtain the consensus value. Finally, a consensus vector can be obtained from each Edge PE in the Edge cloud *E*_1_. Among them, each element in the consensus vector represents the consensus value of each Edge PE in the in the Edge cloud *E*_1_. To calculate the majority value of each element in the consensus vector, the consensus value of the Edge cloud *E*_1_ is obtained. Figure 8e,f shows the consensus values obtained by Edge PEs *e*_12_ and *e*_13_, respectively. Finally, the consensus value (=1) is obtained by each Edge PE in the Edge cloud, and the consensus value is transmitted to the Cloud-layer.

When the Cloud PE in the Cloud-layer receives the consensus values sent by the Edge PEs in the Edge cloud of Edge-layer, the received consensus values are calculated as a majority value (majority (λ,1,1,1,0,1) = 1). The majority value is used as the initial value of Cloud PE to execute OCDAM. Figure 9a shows the initial value of each Cloud PE in the Cloud-layer. In this example, Cloud PE only needs to exchange data twice to execute the Data Gathering Stage (σ = ⌊(*n* − 1)/3⌋ + 1+1 = ⌊(6 − 1)/3⌋ + 1 = 2, where *n_C_* is the number of Cloud PE in the Cloud-layer). Then, OCDAM(σ, *v_i_*, *n_C_*) = OCDAM(2, 1, 6) is executed by Cloud PE.

After that, OCDAM is executed by each Cloud PE in the Cloud-layer. In the first data exchange of the Data Gathering Stage, the initial value of each Cloud PE is transmitted to all other Cloud PEs, and *n_C_* (=6) data received from other *n_C_* Cloud PEs are stored in the level 1 of its corresponding *dg-graph*. The *dg-graph* of each Cloud PE in Cloud-layer at the first time of Data Gathering Stage is shown in Figure 9b. In the second data exchange, each Cloud PE sends the data stored in the level 1 of its *dg-graph* to other Cloud PEs in the Cloud-layer and stores the received data in the *n_C_* (=6) nodes of its *dg-graph*. Figure 9c,d show the *dg-graph**s* established by Cloud PE *c*_2_ and *c*_3_, respectively during Data Gathering Stage.

Subsequently, the function VOTE(α) is applied to level 1 of the *dg-graph* with each Cloud PE to obtain the consensus value in the Consensus Decision Stage. Then, the consensus vector (1,1,1,0,0,1) can be obtained by each Cloud PE in the Cloud-layer. The consensus vectors of Cloud PE *c*_2_ and *c*_3_ are shown in Figure 9e,f, respectively. Finally, through the provision of each Cloud PE in the Cloud-layer, the consensus of the CIoT service constructed by ECIoT can be reached.

## 6. The Correctness and Complexity of the Proposed Method

There are two main ways to solve a problem: Proofs and simulation/experiment. The most complete method is to use mathematical logic to prove the correctness of the solution proposed to solve the problem. When the problem is too sophisticated to derive a mathematical proof, most researchers can use computer simulation to find out the possible solutions or phenomenon [36]. Since the consensus problem is a theoretical problem, most related studies in the past have proved the optimization of the consensus problem through mathematical methods without any experiments [10,11,22,23,24,25,26,27,37]. In the paper, an example with a simple experiment had been shown in Section 5. The pseudo code had been provided in Appendix C for further simulation of the protocol by using any simulation tools. The correctness and complexity of the protocol OCDAM will be proved following the method of [10,11,22,23,24,25,26,27,37] in this section. First, the protocol proposed in this research can guarantee the constraints: Termination, Agreement, and Integrity in Section 6.1. In addition, the optimization of the proposed protocol will be verified by two points: (1) The times of data exchange required to reach a consensus is minimal, and (2) the number of dormant and malicious abnormal PEs that can be allowed is maximal.

The parameters used for the proof of the correctness and complexity of the proposed protocol are listed in detail in Table 3.

### 6.1. The Correctness Verification

To prove the correctness of the proposed protocol, a vertex α is called common if each normal PE has the same value for α [10]. That is, if vertex α is common, then the value stored in vertex α of each normal PE’s *dg-graph* is identical. When each normal PE has a common initial value of *PE_i_* in the root of the *dg-graph*, if the root *nd*(*i*) of the *dg-graph* in a normal PE is common and the initial value received from the *PE_i_* is stored in the root of the *dg-graph*, then the consensus is reached because the root is common. Thus, the constraints (Termination), (Agreement), and (Integrity) can be rewritten as:

**Termination’:** The value of Root *i* can be determined eventually, if the *PEi* is normal.

**Agreement’**: Root *i* is common.

**Integrity’:** VOTE(*i*) = *v_i_* for each normal PE, if the *PE_i_* is normal.

To prove that a vertex is common, the term common frontier is defined as follows: When every root-to-leaf path of the *dg-graph* contains a common vertex, the collection of the common vertices forms a common frontier. In other words, every normal PE has the same data collected in the common frontier if a common frontier does exist in a normal PE’s *dg-graph*; subsequently, using the same majority voting function to compute the root value of *dg-graph*, every normal PE can compute the same root value because the same input (the same collected data in the common frontier) and the same computing function will cause the same output (the root value).

Since the proposed method can solve the consensus problem, the correctness of the proposed method should be examined by the following two terms.
(1)Correct vertex: Vertex *αi* of *dg-graph* is a correct vertex if *PE_i_* (the last PE name in the name list of vertex *αi*) is normal. In other words, a correct vertex is a place to store the value received from a normal PE.(2)True value: For a correct vertex ai in the dg-graph of a normal PE, nd(ai) is the true value of vertex ai. In other words, the stored value for a correct vertex is called the true value.

By the definition of a correct vertex, its stored data is received from the normal PE, and a normal PE always transmits the same data to all PEs; therefore, the correct vertices of such *dg-graph* are common. Thus, the root can be proven a common vertex [(Agreement’) is true] due to the existence of a common frontier, regardless of the correctness of *PE_i_*. The consensus on the root value can now be reached.

Next, the validity of (Integrity’) needs to be checked. When *PE_i_* is abnormal, (Integrity’) is true due to the propositional logic [(P➜Q)] means (NOT(P) OR Q), hence (NOT(P) OR Q) or (P➜Q) is true when P is false, where P implies “*PE_i_* is abnormal” and (P➜Q) implies (Integrity’) [9]. Conversely, root *i* is a correct vertex by the definition of a correct vertex if *PE_i_* is normal. If all the correct vertices’ true values can be computed by the proposed method, then the true value of the root can also be computed because the root is a correct vertex. By definition, the true value of the root is the initial value of *PE_i_* if the *PE_i_* is normal. In short, each normal PE’s root value is the initial value of *PE_i_* if *PE_i_* is normal; therefore, (Integrity’) is true when *PE_i_* is normal.

Meanwhile, the ECIoT network discussed in the study is synchronous, and the protocol OCDAM will stop all normal PEs to exchange data as if the upper bound of times of data exchange is reached. Every normal *PE_i_* executes Consensus Decision Stage to determine VOTE(*i*). The condition (Termination’) is satisfied [38]. Since (Agreement’), (Integrity’) and (Termination’) are true no matter whether *PE_i_* is normal or abnormal, the consensus is solved. 

**Lemma** **1.**
*The data sent by a dormant abnormal PEs can be detected by the normal receiving PEs.*


**Proof.** If the protocol encodes the transmitting messages by the Manchester code, the dormant abnormal PE can be detected by the receiving PE. □

**Theorem** **1.***A normal receiving PE can receive data from sending PEs without influence from any abnormal PEs between the sending PE and receiving PE in same cluster i if n_Bj_* > ⌊(*n_Bj_* − 1)/2⌋ + *f_mBj_* + *f_dBj_ or n_Ej_* >⌊(*n_Ej_* − 1)/3⌋ + 2*f_mEj_* + *f_dEj_ or n_C_* > ⌊(*n_C_* − 1)/3⌋ + 2*f_m__C_* + *f_d__C_*.

**Proof.** By Lemma 1, we can remove the influence of dormant abnormal PEs between any paired sending PE and receiving PE in each time of data exchange, and we can rule out the influence of malicious abnormal PEs between any pairs of PEs in each time of data exchange if *n_Bj_* > ⌊(*n_Bj_* − 1)/2⌋ + *f_mBj_* + *f_dBj_ or n_Ej_* > ⌊(*n_Ej_* − 1)/3⌋ + 2*f_mEj_* + *f_dEj_ or n_C_* > ⌊(*n_C_* − 1)/3⌋ + 2*f_m__C_* + *f_dC_*. This is because the normal sending PE sends *n_Bj_* (or *n_Ej_* or *n_C_*) copies of data to normal receiving PEs. In the worst case, a normal receiving PE receives *n_Bj_ − f_mBj_* + *f_dBj_* (or *n_Ej_ − f_mEj_* + *f_dEj_* or *n_C_* − 2*f_mC_* + *f_dC_*) data transmitted by the normal sending PE because information from dormant abnormal PEs can be detected. Therefore, a normal receiving PE can determine the normal data by taking the majority value. □

**Lemma** **2.**
*A normal receiving PE can detect the dormant abnormal sending PE.*


**Proof.** If the number of λ is greater than or equal to (*n_i_* − 1) − ⌊(*n_i_* − 1)/3⌋ where *n_i_* is the number of PEs in cluster i, then the sending PE has a dormant abnormality. This is because there are at most ⌊(*n_i_* − 1)/3⌋ malicious abnormal PEs in the network, hence there are at most ⌊(*n_i_* − 1)/3⌋ non-λ data. □

**Theorem** **2.**
*A normal PE can detect all dormant abnormal PEs in the ECIoT.*


**Proof.** In the protocol OCDAM, there are ⌊(*n* − 1)/3⌋ + 1 times of data exchanges in cluster i, where *n* ≥ 4. Thus, there are at least two times of data exchanges during the Data Gathering Stage. Each normal PE can receive the data from the cluster *i* during the first time of Data Gathering Stage and receive other PEs’ data during the second time of Data Gathering Stage. Therefore, each PE of cluster *i* can receive all other PEs’ data in the same cluster after two times of data exchanges. According to Lemma 2, each normal PE can detect all dormant abnormal PEs within the cluster. □

**Lemma** **3.**
*All proper vertices of dg-graph are common.*


**Proof.** There are no repeatable vertices remain in dg-graph. At the level ⌊(*n* − 1)/3⌋ + 1 or above, the correct vertex α has at least 2⌊(*n* − 1)/3⌋ + 1 children in which at least ⌊(*n* − 1)/3⌋ + 1 children are correct. The true value of these ⌊(*n* − 1)/3⌋ + 1 correct vertices is in common, and the majority value of vertex α is common. The correct vertex α is common in the *dg-graph*, if the level of α is less than ⌊(*n* − 1)/3⌋ + 1. As a result, all correct vertices of the *dg-graph* are common. □

**Lemma** **4.**
*A common frontier exists in the dg-graph of the normal PE.*


**Proof.** There are ⌊(*n* − 1)/3⌋ + 1 vertices along each root-to-leaf path of the dg-graph in which the root is labeled by the name of PE_i_, and the others are labeled by a sequence of PE names. Since at most ⌊(*n* − 1)/3⌋ PEs can be failed, there are at least one vertex that is correct along each root-to-leaf path of the *dg-graph*. By Lemma 3, the correct vertex is common, and the common frontier exists in each normal PE’s *dg-graph*. □

**Lemma** **5.**
*Let α be a vertex, α is common if there is a common border in the subtree rooted at α.*


**Proof.** If the height of α is 0 and the common border of α exists, then α is common. If the height of α is δ and the children of α are all consensus, by induction, the vertex α is common for the children of height at δ-1. □

**Corollary** **1.**
*The root is common if a common border exists in the dg-graph.*


**Theorem** **3.**
*The root of a normal PE’s dg-graph is common.*


**Proof.** By Lemmas 3–5, and Corollary 1, the theorem is proven. □

**Theorem** **4.**
*The proposed method solves the consensus problem in ECIoT.*


**Proof.** To prove the theorem, it must be shown that the proposed method meets (Termination’), (Agreement’) and (Integrity’). □

(Termination’): According to Theorems 1 and 2, each normal *PE_i_* can receive data from the sending PE without being affected by any abnormal PE after performing the Data Gathering Stage of OCDAM within ⌊(*n* − 1)/3⌋ + 1 times of data exchanges, where *n* ≥ 4. Then, each normal *PE_i_* executes the Consensus Decision Stage of OCDAM to determine VOTE(*i*). Therefore, no more data transits and a value VOTE(*i*) can be decided on, the condition (Termination’) is satisfied.

(Agreement’): By Theorem 3, the root of a normal *PE’s*
*dg-graph* is common; hence, (Agreement’) is satisfied.

(Integrity’): If *PE_i_* is normal, then it broadcasts the same initial data *v_i_* to all PEs. The data of proper vertices for all normal PEs’ *dg-graph* is *v_i_*. Thus, each proper vertex of the *dg-graph* is common (by Lemma 1), and its data are *v_i_*. Since the *PE_i_* is normal, the root of the *dg-graph* is also a proper vertex by Lemma 5. By Theorem 3, this root is common. The computed value VOTE(*i*) = *v_i_* is stored in the root for all normal PEs. Thus, (Integrity’) is satisfied.

### 6.2. The Complexity Verification

The complexity of the proposed method will be verified by two factors: (1) The times of data exchange required, and (2) the total number of abnormal PEs allowed. Theorems 5 and 6 have proved that the proposed method solves the consensus problem by using the minimum times of data exchange and allowing the maximum number of abnormal PEs, respectively. Therefore, the optimality of the proposed method will be obtained.

**Theorem** **5.**
*The times of data exchange required to reach consensus with the proposed method is the minimum.*


**Proof.** In order to obtain the total times of data exchange required by the method proposed in this research, the proof will calculate the times of data exchange required for each layer of ECIoT separately. □

(1)Access-layer: In the Access-layer, each CIoT PE sends the sensed data to the Edge-layer during the Data Gathering Stage. Therefore, only one data exchange is required.(2)Edge-layer: When OCDAM is executed, data exchange is only required during the Data Gathering Stage. According to the research results of [10,11,26], in a distributed system composed of *n* PEs, ⌊(*n* − 1)/3⌋ + 1 is the minimum times of data exchange required to collect enough data to reach a consensus. Because the Edge PEs may be in a dormant or malicious abnormal state in the Edge-layer of ECIoT, each Edge PE in the Edge-layer must exchange data with other Edge PEs to collect enough data to eliminate the influence of abnormal PEs. Therefore, the minimum times of data exchange proposed in [10,11,26] can be applied to the Edge-layer. In other words, in the Edge-layer, there are *n_Ej_* Edge PEs in the Edge cloud *E_j_* of Edge-layer, and OCDAM needs to exchange ⌊(*n_Ej_* − 1)/3⌋ + 1 times of data. In the *E*-cloud Edge-layer, the Edge PE in each Edge cloud executes OCDAM in parallel; hence, the times of data exchange required for each Edge PE to perform OCDAM in all Edge Clouds depends on the number of Edge PEs in the Edge cloud.(3)Cloud-layer: The times of data exchange required to discuss in Cloud-layer is similar to that of Edge-layer discussions. The results of [10,11,26] can still be applied to the Cloud-layer. In the Cloud-layer, there are *n_C_* Cloud PEs, so the Cloud PE needs ⌊(*n_C_* − 1)/3⌋ + 1 times to exchange data when executing the Data Gathering Stage of OCDAM. In other words, when *n_C_* Cloud PE exists in the Cloud-layer, OCDAM will be executed by *n_C_* Cloud PE. At this time, each Cloud PE needs to perform ⌊(*n_C_* − 1)/3⌋ + 1 data exchanges before reaching a consensus.

According to the description, the proposed method requires the minimum times of data exchange when the consensus is reached.

**Theorem** **6.**
*The total number of abnormal PEs allowed by OCDAM is the maximum.*


**Proof.** In this proof, the total number of abnormal PEs allowed by OCDAM will be discussed separately through the three layers of ECIoT. □

(1)Access-layer: Since the number of abnormal CIoT PEs within the communication range of each specific BS in the Access-layer cannot exceed half, otherwise no consensus can be reached. According to the research result of Babaoglu and Drummond [23], *n_Bj_* > ⌊(*n_Bj_* − 1)/2⌋ + *f_mBj_* + *f_dBj_* can be used to describe the number of CIoT PEs required in the communication range of a specific *BS_j_* at the Access-layer. Then, *F_A_* is defined as the total number of dormant and malicious abnormal PEs allowed in the Access-layer, *F_A_* = $\sum_{j=1}^{B}f_{Bj}$ and *f_Bj_* = *f_mBj_* + *f_dBj_*, where *B* is the total number of BSs in the Access-layer. In addition, *n_Bj_* > ⌊(*n_Bj_* − 1)/2⌋ + *f_mBj_* + *f_dBj_* is used to describe the number of CIoT PEs required in the coverage of a specific *BS_j_* at the Access-layer.(2)Edge-layer: According to the research results of Wang et al. [26], in a distributed computing system with *n* PEs, the condition for reaching a consensus problem is *n* > ⌊(*n* − 1)/3⌋ + 2*f_m_* + *f_d_*. Since ECIoT is a distributed computing system, the research results of Wang et al. [26] can be directly applied to the Edge-layer. Therefore, in Edge cloud *E_j_* at the Edge-layer, the result that can be obtained is *n_Ej_* >⌊(*n_Ej_* − 1)/3⌋ + 2*f_mEj_* + *f_dE_*. Then, *F_E_* = $\sum_{j=1}^{E}f_{Ej}$ and *f_Ej_* = *f_mEj_* + *f_dEj_*, where *E* is the total number of Edge clouds in the Edge-layer. Furthermore, *n_Ej_* > ⌊(*n_Ej_* − 1)/3⌋ + 2*f_mEj_* + *f_dEj_* is used to describe the number of Edge PEs in the Edge cloud *E_j_* at the Edge-layer.(3)Cloud-layer: The same as calculating the number of abnormal Edge PEs allowed in Edge-layer, the results of Wang et al. [26] can also be directly applied to the Cloud-layer. In a Cloud-layer composed of *n_C_* Cloud PEs, *n_C_* > ⌊(*n_C_* − 1)/3⌋ + 2*f_mC_* + *f_dC_* can be obtained. Then, *F_C_* = *f_mC_* + *f_dC_* is the total number of abnormal PEs allowed in the Cloud-layer, *n_C_* > ⌊(*n_C_* − 1)/3⌋ + 2*f_mC_* + *f_dC_* is used to describe the number of Cloud PEs required in the Cloud-layer.

By adding the allowable number of abnormal PEs in the three layers of ECIoT, *F* = *F_A_* + *F_E_* + *F_C_* = $\sum_{j=1}^{B}f_{Bj}$ + $\sum_{j=1}^{E}f_{Ej}$ + *F_C_*, then the maximum number of abnormal PEs allowed by the proposed method can be obtained. In other words, *F* is the maximum number of abnormal PEs allowed by executing the proposed method in ECIoT to reach consensus.

Through the proofs in this section, the method proposed in this study can guarantee the three constraints for solving the consensus problem, including Termination, Agreement, and Integrity. The proposed method can be done with the minimum times of data exchange and can tolerate the maximum number of dormant and malicious abnormal PEs, so that normal PEs can reach a consensus. Therefore, the correctness and complexity of the proposed method is proved.

## 7. Conclusions and Future Works

In this section, the conclusion of our research and the future works will be discussed.

### 7.1. Conclusion of Our Research

The IoT is the most viable technology to achieve connected life. Pervasive connectivity can be achieved through intelligent, automatic, and perceptual physical objects that can think and act intelligently without human intervention. Through the use of the IoT, it is expected that the cost of personnel and organizations can be reduced, and a variety of novel applications can be provided at the same time [2]. Wireless communication is one of the most successful technologies in recent years. Through wireless communication, the complexity associated with the IoT can be managed. It provides many potentially destructive elements for traditional people-oriented broadband networks [39]. In addition, the cellular network is expected to increase capacity, reduce end-to-end delay, improve reliability, and increase coverage, and may even meet the most demanding IoT requirements [40].

The architecture of edge computing is the latest enhancement of network processing capabilities, in which computing/storage capabilities are placed near the end user [35]. Therefore, the cellular network that provides CIoT services and provides the functions of edge computing is very suitable for the widespread applications of IoT in the future. In order to provide highly reliable services to these applications, a highly reliable CIoT environment is required to support these large-scale applications. Consequently, the consensus problem can achieve this goal.

The consensus problem is one of the important issues discussed to improve the reliability of distributed systems. Among them, the topology of the network is one of the important factors that affect the resolution of consensus problems. In this study, ECIoT is a CIoT platform integrated with edge computing to improve the high-quality services of CIoT. In this research, the proposed method is used to solve the consensus problem that PEs may be dormant or malicious abnormal in ECIoT. Since the consensus problem is a theoretical problem, many related studies in the past have proved its optimization through mathematic methods without conducting any experiments [10,11,22,23,24,25,26,27,38]. Therefore, detailed proofs have been shown in this study to verify the optimization of the proposed method.

Due to the difference in network topology, the implementation of the consensus will be affected, and all the protocols related to the consensus in the past are not suitable for use under the topology of ECIoT. Therefore, in order to improve the reliability of ECIoT, the OCDAM protocol is proposed in this study to solve the consensus problem in ECIoT. The state in which the consensus problem has been resolved under different network topologies is shown in Table 1. From Table 1, Meyer and Pradhan [11] focused on FCN, Wang et al. [25] focused on MCN, Wang et al. [26] focused on IFIoT, and OCDAM focused on ECIoT is proposed in this study to discuss the case of the dual abnormality mode, while other related studies only discussed malicious anomalies.

The protocol CIoTAP proposed by Pan and Wang [27] can only tolerate malicious abnormal PEs in an ECIoT and the maximum number of abnormal PEs allowed is $\sum_{j=1}^{B}f_{mBj}$ + $\sum_{j=1}^{E}f_{mEj}$ + *f_mC_*. In this research, the proposed protocol OCDAM can tolerate both dormant and malicious abnormal PEs existing simultaneously in ECIoT and the maximum number of abnormal PEs allowed is $\sum_{j=1}^{B}f_{mBj}+f_{dBj}$ + $\sum_{j=1}^{E}f_{mEj}+f_{dEj}$ + *f_mC_* + *f_d_*. Therefore, the fault tolerance capability of our proposed protocol is much better than Pan et al. [27]. Furthermore, Pan et al. [27] lacked the proof of the correctness of their protocol. Conversely, the correctness and the optimality of OCDAM had been both proved in the paper. Based on the proof of this research in Section 6, the proposed consensus protocol OCDAM can indeed use the minimum times of data exchange to ensure that all normal PEs in ECIoT can reach a consensus. Meanwhile, OCDAM can allow the maximum number of dormant and malicious abnormal PEs existed in ECIoT. To sum up, the protocol OCDAM is optimal to make all normal PEs reach Termination, Agreement, and Integrity underlying ECIoT.

On the other hand, a simple instance is shown by using OCDAM in ECIoT, and a highly reliable IoT application can be built. Because ECIoT is a distributed computing system built by integrating edge computing, ECIoT can be widely used in the design and practice of various distributed computing systems to provide the relevant CIoT services required by users.

### 7.2. Future Works

The fallible component is restricted to abnormal PEs in the paper. It is not enough to realize a highly reliable ECIoT. In the more generalized ECIoT, not only PEs may be fallible in the network, but also transmission media may be fallible [30,41,42]. Therefore, in future research, when the abnormal PEs and transmission media both exist in ECIoT simultaneously, the proposed protocol will be extended to solve the generalized consensus problem.

In addition, in order to maintain the reliability of ECIoT, another related problem called the Fault Diagnosis Protocol (FDA) [43,44,45,46] will be discussed in the future. If a protocol can be proposed to help each PE detect and locate abnormal components in ECIoT, then the reliability of ECIoT can be maintained to provide a stable application service environment for CIoT.

## Figures and Tables

**Figure 1 sensors-21-00671-f001:**
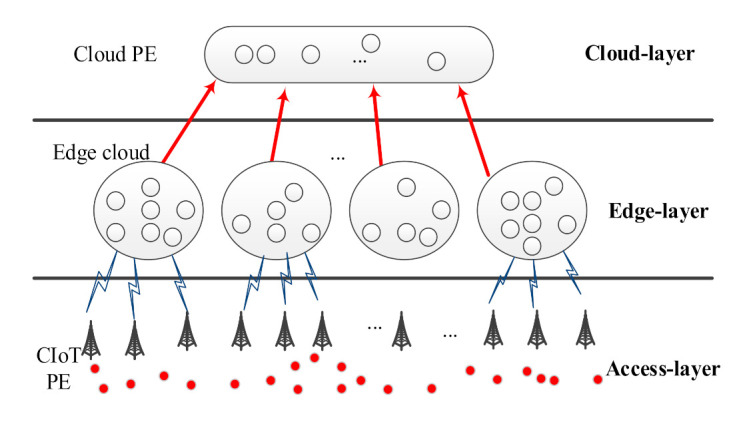
The structure of ECIoT.

**Figure 2 sensors-21-00671-f002:**
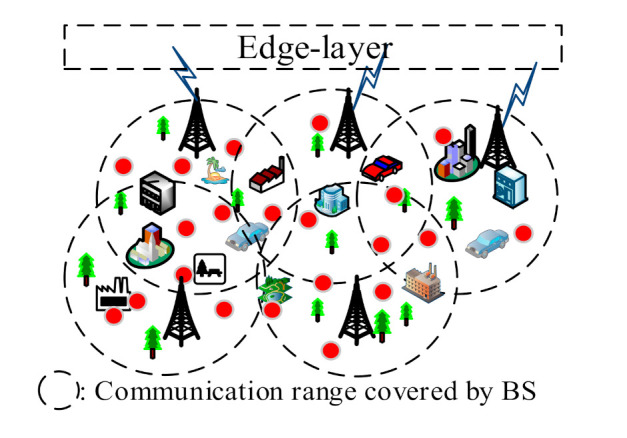
The Access-layer of ECIoT.

**Figure 3 sensors-21-00671-f003:**
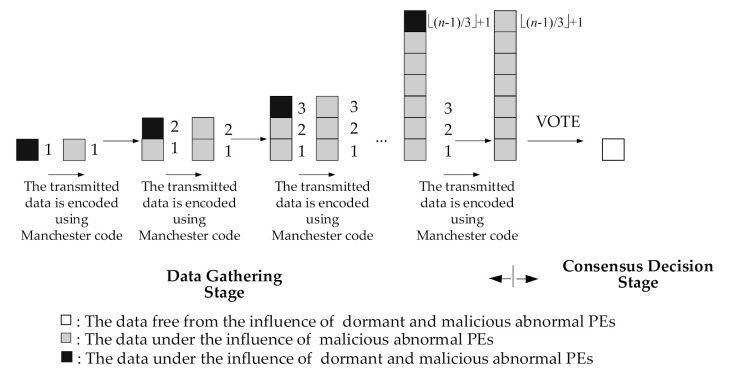
The progression of the influence of dormant and malicious abnormal processing elements (PEs) removed.

**Figure 4 sensors-21-00671-f004:**
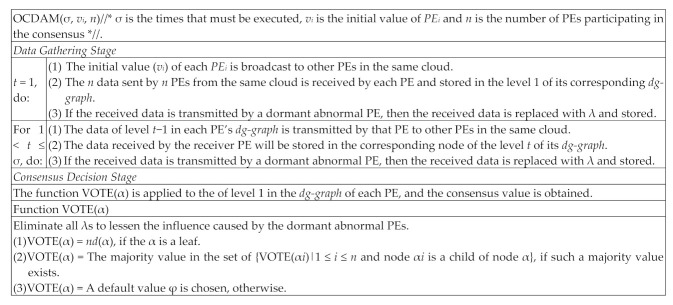
The proposed OCDAM.

**Figure 5 sensors-21-00671-f005:**
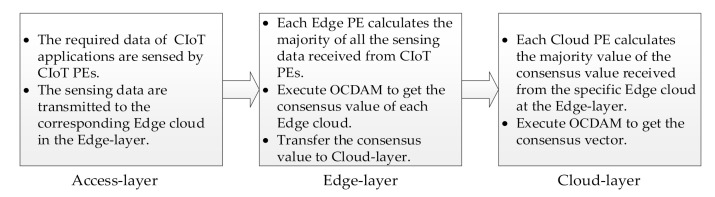
The execution steps of the proposed method.

**Figure 6 sensors-21-00671-f006:**
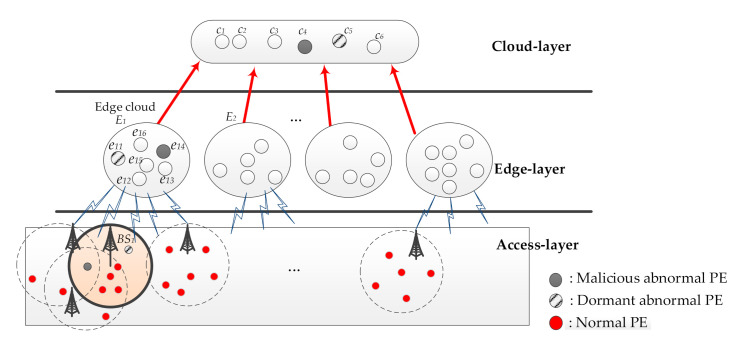
The example environment constructed by ECIoT.

**Figure 7 sensors-21-00671-f007:**
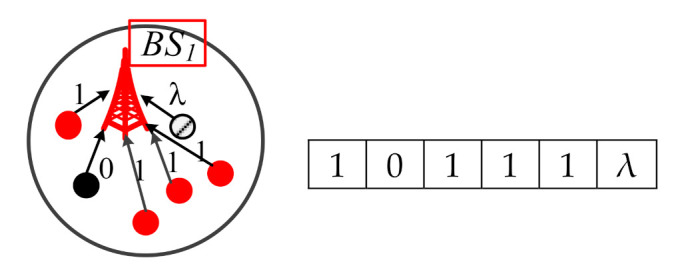
An example of the communication range of a specific *BS*_1_.

**Figure 8 sensors-21-00671-f008:**
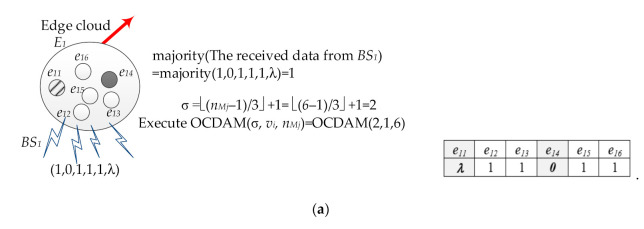
(**a**) The initial value of each PE in Edge cloud *E*_1_ of Edge-layer. (**b**) The *dg-graph* of each PE in Edge cloud *E*_1_ during the first data exchange in the Data Gathering Stage. (**c**) The final *dg-graph* of *e*_12_ during the second data exchange in the Data Gathering Stage. (**d**) The final *dg-graph* of *e*_13_ during the second data exchange in the Data Gathering Stage. (**e**) The consensus value of e12 by Consensus Decision Stage. (**f**) The consensus value of e13 by Consensus Decision Stage.

**Figure 9 sensors-21-00671-f009:**
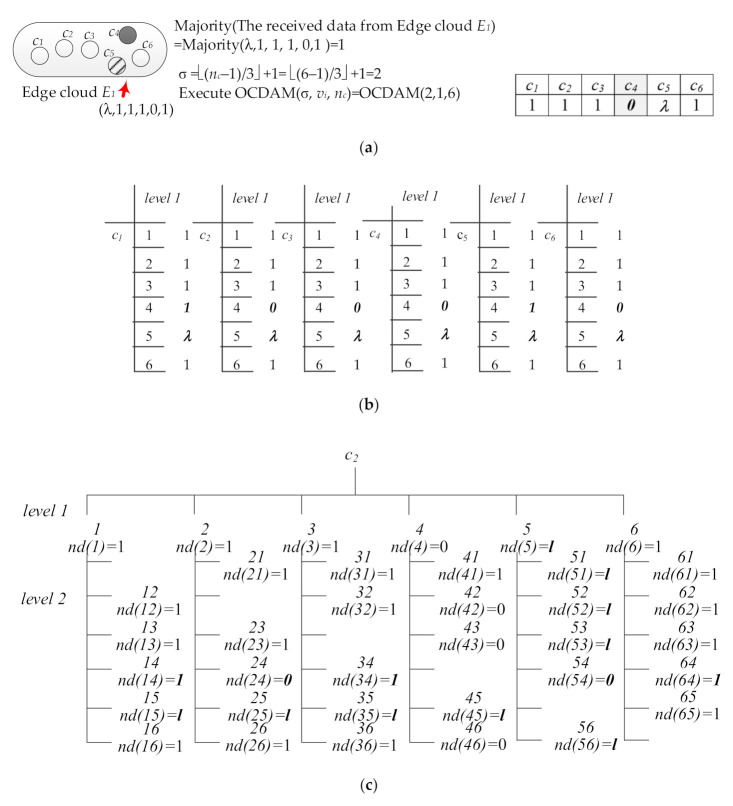
(**a**) The initial value of each Cloud PE of Cloud-layer. (**b**) The *dg-graph* of each Cloud PE in Cloud-layer during the first data exchange in the Data Gathering Stage. (**c**) The final *dg-graph* of *c*_2_ during the second data exchange in the Data Gathering Stage. (**d**) The final *dg-graph* of *c*_3_ during the second data exchange in the Data Gathering Stage. (**e**) The consensus vector of *c*_2_ by Consensus Decision Stage. (**f**) The consensus vector of *c*_3_ by Consensus Decision Stage.

**Table 1 sensors-21-00671-t001:** The comparison of previous various protocols over different network models.

	Topology	BCN		FCN		GCN		MCN		CC		IFIoT		ECIoT	
Results		*da*	*ma*	*da*	*ma*	*da*	*ma*	*da*	*ma*	*da*	*ma*	*da*	*ma*	*da*	*ma*
Babaoglu & Drummond [23]			V												
Fischer & Lynch [10]					V										
Lamport, Shostak & Pease [22]					V										
Meyer & Pradhan [11]				V	V										
Wang, Chin & Yan [24]							V								
Wang, Yan & Cheng [25]								V	V						
Beheshti & Safi-Esfahani [34]											V				
Wang, Tseng, Yan & Tsai [26]												V	V		
Pan & Wang [27]															V
OCDAM														V	V

**Table 2 sensors-21-00671-t002:** The comparison of the proposed protocol Optimal Consensus with Dual Abnormality Mode (OCDAM) and CIoTAP in the edge-computing-based cellular Internet of Things (ECIoT).

*n*		6			7				8			
OCDAM	*m*	0	1	2	0	1	2	3	0	1	2	3
	*d*	≤5	≤3	≤1	≤6	≤4	≤2	≤0	≤7	≤5	≤3	≤1
CIoTAP [27]	*m*	0	1	2	0	1	2	3	0	1	2	3
	*d*	0	0	0	0	0	0	0	0	0	0	0

**Table 3 sensors-21-00671-t003:** The parameters used in optimization proof.

Parameter	Meaning
*BS_j_*	the base station *j* at the Access-layer
*n_Bj_*	the total number of CIoT PEs within the communication range of *BS_j_*
*f_mBj_*	the total number of allowable malicious abnormality PEs within the communication range of *BS_j_*
*f_dBj_*	the total number of allowable dormant abnormality PEs within the communication range of *BS_j_*
*f_Bj_*	the total number of abnormal CIoT PEs allowed in the communication range of *BS_j_* and *f_Bj_* = *f_mBj_* + *f_dBj_*
*F_A_*	the total number of allowable dormant and malicious abnormal PEs in Access-layer
*E_j_*	Edge cloud *j* at the Edge-layer
*n_Ej_*	the total number of Edge PEs in Edge cloud *E_j_*
*f_mEj_*	the total number of allowed malicious abnormal Edge PEs in Edge cloud *E_j_*
*f_dEj_*	the total number of allowed dormant abnormal Edge PEs in Edge cloud *E_j_*
*f_Ej_*	the total number of abnormal Edge PEs allowed in Edge cloud *E_j_* and *f_Ej_* = *f_mEj_* + *f_dEj_*
*F_E_*	the total number of allowed dormant and malicious abnormal PEs in the Edge-layer
*n_C_*	the total number of Cloud PEs in Cloud-layer
*f_mC_*	the total number of allowed malicious abnormal Cloud PEs in Cloud-layer
*f_dC_*	the total number of allowed dormant abnormal Cloud PEs in Cloud-layer
*F_C_*	the total allowed number of dormant and malicious abnormal PEs in the Cloud-layer and *F_C_* = *f_mC_* + *f_dC_*
*F*	the maximum number of dormant and malicious abnormal PEs allowed by executing OCDAM and *F* = *F_A_* + *F_E_* + *F_C_*

## Data Availability

The data presented in this study are available on request from the corresponding author.

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
