# Peer review of "Optimal Consensus with Dual Abnormality Mode of Cellular IoT Based on Edge Computing"

_sensors, 2021, doi:10.3390/s21020671_

Round 1

Reviewer 1 Report

The paper proposes a consensus algorithm in presence of both malicious peers and crashed/disconnected peers in 5G and IoT networks, viz. in hierarchical edge-cloud settings. Despite tackling a potentially interesting problem, the paper shows some drawbacks which would require a major revision to be fixed:

  1. The current introduction does not motivate enough to the considered problem. Why is it needed to solve the consensus problem in 5G/IoT networks? Is there any practical usage example? How does consensus links to overall system performance? A consensus algorithm must guarantee termination, integrity and agreement; however, only integrity is described in the introduction. Finally, the contribution is not very clearly framed in the introduction.
  2. The distributed consensus problem has been thoroughly studied since di beginning of the '80s and more than 12 different consensus protocols exist, also tolerating crashes and byzantine failures of the network. Surprisingly enough, apart from a brief table and discussion in the conclusions, the paper does not provide any detailed comparison section with related works on this topic, e.g. those in the surveys:
    1. "Recent advances in consensus of multi-agent systems: A brief survey." IEEE Transactions on Industrial Electronics 64.6 (2016): 4972-4983.
    2. Carrara, Gabriel R., et al. "Consistency, availability, and partition tolerance in blockchain: a survey on the consensus mechanism over peer-to-peer networking." Annals of Telecommunications (2020): 1-12.

Adding a Related Work section is needed also to better evaluate the novelty of the proposed solution with respect to the state of the art.

3. No implementation (for actual testbeds or simulation purposes) of the proposed protocol is released, which would enable experimenting with the proposed consensus algorithm. The Reviewer would strongly recommend implementing and open-sourcing the proposed solution to run further experiments in either testbed or simulation environments.

3. Assessment is carried over a motivating example and is very difficult to follow and understand for the readers as some of the Figures 8-9 are too dense. Last, the Reviewer wonders whether all 19 parameters in Table 2 are actually needed.

4. Very few lines for research work are highlighted in the Conclusions. The Authors should try to expand on this.

Author Response

Response to Reviewer 1 Comments

Point 1: The current introduction does not motivate enough to the considered problem. Why is it needed to solve the consensus problem in 5G/IoT networks? Is there any practical usage example? How does consensus links to overall system performance? A consensus algorithm must guarantee termination, integrity and agreement; however, only integrity is described in the introduction. Finally, the contribution is not very clearly framed in the introduction.

Response 1:

(1) Thank you for your comments and kindness to read our paper.

(2) With the 5G network will greatly expand the application of the IoT, thereby promoting the operation of cellular networks, the security and network challenges of the IoT, and pushing the future of the Internet to the edge [1]. Federated learning (FL) is a model of machine learning in distributed systems. In the research of Savazzi et al. [12], the proposed FL algorithms leverage the cooperation of devices that perform data operations inside the network by iterating local computations and mutual interactions via consensus-based methods. This method lays the foundation for the integration of FL into 5G and networks characterized by decentralized connectivity and computing, as well as intelligence integrated on edge devices. In the study of Lin et al. [13], a practical collaboration infrastructure for 5G network slice broker is designed, where the core challenge is the consensus protocol to guarantee the security and performance of the overall system. In the other words, by solving the consensus problem, many related applications can be realized, such as the adaptive weighted replication [14,15], information retrieval [16,17], and the flight control system [18,19]. In addition, the consensus problem has also been studied and widely used in various fields such as blockchain and IoT [20,21]. The descriptions are shown in lines 58-71.

(3) The contribution of this research is: “This research solves the consensus problem of PE in dual abnormality mode, in which both dormant abnormal PEs and malicious abnormal PEs are existed simultaneously in the system. In these cases, the reliability of the system will be maximized. The protocol proposed in this research, Optimal Consensus with Dual Abnormality Mode (OCDAM), can make all normal PEs reach a consensus on the minimum times of data exchange and tolerate the maximum number of abnormal PEs.” And, the contribution of this research is shown in lines 102-110.

Point 2: The distributed consensus problem has been thoroughly studied since di beginning of the '80s and more than 12 different consensus protocols exist, also tolerating crashes and byzantine failures of the network. Surprisingly enough, apart from a brief table and discussion in the conclusions, the paper does not provide any detailed comparison section with related works on this topic, e.g. those in the surveys:

  1. "Recent advances in consensus of multi-agent systems: A brief survey." IEEE Transactions on Industrial Electronics 6 (2016): 4972-4983.
  2. Carrara, Gabriel R., et al. "Consistency, availability, and partition tolerance in blockchain: a survey on the consensus mechanism over peer-to-peer networking." Annals of Telecommunications (2020): 1-12.

Adding a Related Work section is needed also to better evaluate the novelty of the proposed solution with respect to the state of the art.

Response 2:

In Section 2.2, we use following statements to explain the focus of our briefly review: “Because the solution of consensus problems is one of the most commonly used methods in the field of providing reliable distributed systems, many protocols have previously been proposed to solve the consensus problem for different application areas, such as multi-agent systems, peer-to-peer networks [26,27]. In this study, we focus on the basic protocol of reaching consensus underlying different network topologies. In previous results of this field, the consensus problem was solved in many network models with various fallible component assumptions, such as a BroadCasting Network (BCN) [28], a Fully Connected Network (FCN), a Generalize Connected Network (GCN) [29], a MultiCasting Network (MCN) [30], a Cloud Computing environment (CC) [31], an Integrated Fog IoT (IFIoT) [32], and an Edge computing-based CIoT (ECIoT) [24].” A comparison and review of previous consensus protocols in different network topologies has been provided in the Section 2.2. The full descriptions are shown in lines 162-190.

Point 3: No implementation (for actual testbeds or simulation purposes) of the proposed protocol is released, which would enable experimenting with the proposed consensus algorithm. The Reviewer would strongly recommend implementing and open-sourcing the proposed solution to run further experiments in either testbed or simulation environments.

Response 3:

Since the consensus problem is a theoretical problem, many related studies in the past have proved the optimization of the consensus problem through mathematical methods without any experiments [9,11,24,28-32]. Therefore, the optimization of the proposed method will be explained in Section 5. In the proof, the optimization of the proposed method will be verified by the minimum times of data exchange required to reach a consensus and the maximum number of dormant and malicious abnormal PEs that can be allowed when a consensus is reached. Through these two proofs, the proposed method will prove to be the optimal solution to the consensus problem under the ECIoT topology. The descriptions are shown in lines 414-421.

Point 4: Assessment is carried over a motivating example and is very difficult to follow and understand for the readers as some of the Figures 8-9 are too dense. Last, the Reviewer wonders whether all 19 parameters in Table 2 are actually needed.

Response 4:

  1. All figures in this article have been readjusted.
  2. In order to focus on the proof of the optimization of the proposed method, the parameters used in the proof are listed in Table 3. In order to simplify Table 3, the parameters that did not affect the proof have been deleted in accordance with the comments of the reviewer.

Point 5: Very few lines for research work are highlighted in the Conclusions. The Authors should try to expand on this.

Response 5:

Due to the difference in network topology, the implementation of the consensus will be affected, and all the protocols related to the consensus in the past are not suitable to use under the topology of ECIoT. Therefore, in order to improve the reliability of ECIoT, the OCDAM protocol is proposed for the first time in this study to solve the consensus problem in ECIoT. The state in which the consensus problem has been resolved under different network topologies is shown in Table 2. From Table 2, Meyer and Pradhan [11] focused in FCN, Wang et al. [30] focused in MCN, Wang et al. [32] focused in IFIoT, and OCDAM focused in ECIoT is proposed in this study to discuss the case of the dual abnormality mode, while other related studies only discussed malicious anomalies. In an ECIoT, Pan & Wang [24] and this research both proposed protocols to solve the consensus problem. However, the protocol proposed by Pan & Wang [24] can only tolerate malicious abnormal PEs. In this research, the proposed protocol can tolerate both dormant and malicious abnormal PEs existed simultaneously in ECIoT. Therefore, the fault tolerance of our proposed protocol is much better than Pan & Wang [24].

In this study, an actual AIoT instance shows that by using OCDAM in ECIoT, a highly reliable AIoT can be built. Because ECIoT is a distributed computing system built by integrating edge computing, ECIoT can be widely used in the design and practice of various distributed computing systems to provide the relevant CIoT services required by users. In short, this research solves the consensus problem of PE in dual abnormality mode that combines dormant abnormality and malicious abnormality, that is, there may be both dormant abnormal PEs and malicious abnormal PEs in the system. According to the proof of this research in Section 5, the proposed consensus protocol OCDAM can indeed use the minimum times of data exchange to ensure that all normal PEs in ECIoT can reach a consensus. In addition, OCDAM can allow the maximum number of dormant and malicious abnormal PEs in ECIoT.

The statement has been presented in lines 516 to 534.

Reviewer 2 Report

This paper proposes OCDAM to solve the consensus problem in the ECIoT environment. The proposed scheme solves both dormant abnormal and malicious abnormal. The dormant abnormal is solved using Manchester code in the data gathering stage, and the function VOTE is used in the consensus decision stage. Performance evaluation verified the optimization of minimum times of data exchange and maximum number of dormant and malicious abnormal PEs through mathematical methods.

Pros

  • In the ECIoT environment, not only malicious abnormalities but also dormant abnormalities are resolved. 
  • The motivation and approach to address the problem looks sound and solid.
  • Section 4 (The Example of the Proposed Method) is helpful to the readers.

Cons to be modified 

  • This paper is basically well-written, but the readability should be improved. There is no overall overview of the scheme at the beginning of Section 3, so it is difficult to understand the overall operation flow easily. Add overview of the proposed scheme at the beginning of Section 3. 
  • In Section 3, The Optimal Consensus with Dual Abnormality Mode (OCDAM) should be described in more detail. (It had better move the detailed explanation in Section 4 to Section 3, and leave just example in Section 4.)
  • The optimization of minimum times of data exchange and maximum number of dormant and malicious abnormal PEs was verified through a mathematical method, but it is necessary to verify how the performance improved in detail compared with other existing studies (especially for malicious abnormal) through experiments.
  • The nodes marked on the access-layer in Figure 6 should be written the same as the text description.
  • Most of related works are out of date. Add explanation and comparison of the latest related research (including schemes for solving consensus problems)

Author Response

Response to Reviewer 2 Comments

Point 1: This paper is basically well-written, but the readability should be improved. There is no overall overview of the scheme at the beginning of Section 3, so it is difficult to understand the overall operation flow easily. Add overview of the proposed scheme at the beginning of Section 3.

Response 1:

An overview of the proposed scheme has been modified at the beginning of Section 3 as following.

“Basically, the principle of OCDAM is to exchange data with each other PEs at first, then to remove the influence of abnormal PEs by taking the majority of data received from other PEs. As if the lower bound of the number of data exchange is completed, all the influences of abnormal PEs are proven to be removed and then the consensus can be reached.

Underlying the ECIoT, the PE of the CIoT is used to sense and transmit required sensing signals to the related application services. The sensed data are sent to the corresponding Edge cloud in the Edge-layer by PEs of the CIoT. Edge PE located in the Edge cloud receives the sensing data sent from PEs of the CIoT, and then the majority value of the received sensing data is obtained. The majority value of the received data is used as the initial value (vi) of the Edge PEi, which will be used to execute OCDAM. When the consensus value of each Edge cloud is obtained, the value is expressed as the result of a specific service. Finally, the consensus value is transmitted to the Cloud-layer by Edge PEs. In ECIoT, Cloud PE is responsible for collecting the results of different specific services in the Cloud-layer, and then the consensus values can be composed to provide an integrated service center for various CIoT applications.”

It has been presented in lines 201 to 214.

Point 2: In Section 3, The Optimal Consensus with Dual Abnormality Mode (OCDAM) should be described in more detail. (It had better move the detailed explanation in Section 4 to Section 3, and leave just example in Section 4.

Response 2:

The detailed explanation OCDAM has been modified and moved from Section 4 to Section 3, and the example is leaved in Section 4. It has been presented in lines 289 to 311.

Point 3: The optimization of minimum times of data exchange and maximum number of dormant and malicious abnormal PEs was verified through a mathematical method, but it is necessary to verify how the performance improved in detail compared with other existing studies (especially for malicious abnormal) through experiments.

Response 3:

Most past researches proposed various protocols to solve the consensus problem underlying various network structures and applications. All of them could not be replaced or compared with each other because the network topology and elements of the network are quite different. In other words, the network structure of ECIoT is more complicated than before; the protocol of reaching consensus underlying ECIoT should be restarted.

The descriptions of “Because the consensus problem is a theoretical problem, most related studies in the past have proved the optimization of the consensus problem through mathematical methods without any experiments [9,11,24,28-32]. Therefore, the optimization of the proposed method will be explained in this paper. In the proof, the optimization of the proposed method will be verified by the minimum times of data exchange required to reach a consensus and the maximum number of dormant and malicious abnormal PEs that can be allowed when a consensus is reached. With these two proofs, the proposed method will be proved to be the optimal solution to the consensus problem under the ECIoT topology.” are shown in lines 414-421.

Point 4: The nodes marked on the access-layer in Figure 6 should be written the same as the text description.

Response 4:

The nodes marked on the access layer in Figure 6 have been corrected.

Point 5: Most of related works are out of date. Add explanation and comparison of the latest related research (including schemes for solving consensus problems)

Response 5:

Most of the related researches the paper referred are published between 2019 and 2020 as following,

A. The related works of 5G and CIoT:

3. Marques, G.; Pitarma, R.; M Garcia, N.; Pombo, N. Internet of Things architectures, technologies, applications, challenges, and future directions for enhanced living environments and healthcare systems: a review. Electronics 2019, 8(10), 1081.
4. Yu, G.; Chen, X.; Ng, D.W.K. Low-cost design of massive access for cellular Internet of Things. IEEE Trans. Commun. 2019, 67(11), 8008-8020.
5. Qi, Q.; Chen, X.; Lei, L.; Zhong, C.; Zhang, Z. Outage-constrained robust design for sustainable B5G cellular internet of things. IEEE Trans. Wirel. Commun. 2019, 18(12), 5780-5790.
6. Lin, Z.N.; Yang, S.R.; Lin, P. Edge computing-enhanced uplink scheduling for energy-constrained cellular internet of things. In 2019 15th International Wireless Communications & Mobile Computing Conference, Tangier, Morocco, Morocco, 24-28 June 2019; IEEE: New York, NY, USA, 2019; pp. 1391-1396.
7. Vukobratovic, D.; Bajovic, D.; Anoh, K.; Adebisi, B. Distributed energy trading via cellular internet of things and mobile edge computing. In IEEE International Conference on Communications, Shanghai, Shanghai, China, 20-24 May 2019; IEEE: New York, NY, USA, 2019; pp. 1-7.
8. Zhang, H.; Di, B.; Bian, K.; Song, L. IoT-U: cellular internet-of-things networks over unlicensed spectrum. IEEE Trans. Wirel. Commun. 2019, 18(5), 2477-2492.
12. Savazzi, S.; Nicoli, M.; Rampa, V. Federated learning with cooperating devices: A consensus approach for massive IoT networks. IEEE Internet Things J. 2020, 7(5), 4641-4654.
13. Lin, W.; Xu, X.; Qi, L.; Zhang, X.; Dou, W.; Khosravi, M.R. A Proof-of-Majority consensus protocol for blockchain-enabled collaboration infrastructure of 5g network slice brokers. In 2nd ACM International Symposium on Blockchain and Secure Critical Infrastructure, Taipei, Taiwan, 5 Oct. 2020; ACM: New York, NY, USA,2020; pp. 41-52.
15. Berger, C.; Reiser, H.P.; Sousa, J.; Bessani, A.N. AWARE: Adaptive wide-area replication for fast and resilient Byzantine consensus. IEEE Trans. Dependable Secur. Comput. 2020, (Early Access).
19. Zhang, X.; Zhao, X. Architecture design of distributed redundant flight control computer based on time-triggered buses for UAVs. IEEE Sens. J. 2020, (Early Access).
23. Badea, A.; Halunga, S.; Berceanu, M.; Găină, M.; Capotă, C.; Stancu, E. Influence of Manchester encoding over spreading codes used in multiple access techniques for IoT purposes. In 2019 IEEE 25th International Symposium for Design and Technology in Electronic Packaging, Cluj-Napoca, Romania, Romania, 23-26 Oct. 2019; IEEE: New York, NY, USA, 2019; pp. 216-219.
25. Chen, X. Massive access for cellular internet of things theory and technique; Springer: Berlin, Germany, 2019.
33. Wanga, S.C.; Lina, W.L.; Hsiehb, C.H. To improve the production of agricultural using IoT-based aquaponics system. Int. J. Appl. Sci. Eng. 2020, 17, 207–222.
34. Agiwal, M.; Saxena, N.; Roy, A. Towards connected living: 5G enabled Internet of Things (IoT). IETE Tech. Rev. 2019, 36(2), 190-202.
35. Hu, J.; Zhang, H.; Song, L.; Han, Z.; Poor, H.V. Reinforcement learning for a cellular internet of UAVs: protocol design, trajectory control, and resource management. IEEE Trans. Wirel. Commun. 2020, 27(1), 116-123.

B. The related works of consensus problems:
14. Berger, C.; Reiser, H.P.; Sousa, J.; Bessani, A. Resilient wide-area Byzantine consensus using adaptive weighted replication. In 2019 38th Symposium on Reliable Distributed Systems, Lyon, France, France, 1-4 Oct. 2019; IEEE: New York, NY, USA, 2019; pp. 183-18309.

18. Sakic, E.; Deric, N.; Goshi, E.; Kellerer, W. P4BFT: Hardware-accelerated Byzantine-resilient network control plane. In 2019 IEEE Global Communications Conference, Waikoloa, HI, USA, 9-13 Dec. 2019; IEEE: New York, NY, USA, 2019; pp. 1-7.
20. Gramoli, V. From blockchain consensus back to byzantine consensus. Futur. Gener. Comp. Syst. 2020, 107, 760-769.
21. Hu, W.; Hu, Y.; Yao, W.; Li, H. A blockchain-based Byzantine consensus algorithm for information authentication of the Internet of vehicles. IEEE Access 2019, 7, 139703-139711.
26. Carrara, G.R.; Burle, L.M.; Medeiros, D.S.; de Albuquerque, C.V.N.; Mattos, D.M. Consistency, availability, and partition tolerance in blockchain: a survey on the consensus mechanism over peer-to-peer networking. Ann. Telecommun. 2020,1-12.

C. The schemes for solving consensus problems
24. Pan, S.H.; Wang, S.C. Enhancing the reliability of cellular internet of things through agreement. Appl. Sci. 2020, 10(21), 7699.

31. Beheshti, M.K.; Safi-Esfahani, F. FPF-Cloud: Applying SVM for Byzantine failure prediction to increase availability and failure tolerance in cloud computing. SN Comput. Sci. 2020, 1(5), 1-31.
32. Wang, S.C.; Tseng, S.C.; Yan, K.Q.; Tsai, Y.T. Reaching agreement in an integrated fog cloud IoT. IEEE Access 2018, 6(1), 64515-64524.

Round 2

Reviewer 1 Report

The Reviewer would thank the Authors for addressing some of the provided comments. However, the Reviewer believes that the manuscript still requires considering some important points in a major revision:

  1. The introduction should be further revised and – possibly – shortened in order to motivate more to the considered problem. A consensus algorithm must guarantee termination, integrity and agreement; however, only integrity is described in the introduction [see https://en.wikipedia.org/wiki/Consensus_(computer_science)]. The Authors should add also termination and agreement, described as separate properties, and relate them to their contribution.
  2. Background on the considered technologies and network structure would be better separated from Related Work. The Authors should consider creating a “Background” and a “Related Work” sections with the material of Sect. 2. While performing this, the related work section should also be extended to highlight the elements of novelty of the proposed approach with respect to the state of the art, e.g., “compared to [X], OCDAM improves on/differs since/…”.
  3. With their theoretical contribution, the Authors are addressing problems in a very practical application scenario from 5G networks. For these reasons, to follow a sound scientific methodology (proofs + experiments), the Reviewer strongly believes that their contribution needs to include an implementation of their solution and experimental assessment against meaningful KPI, at least through simulation (e.g. by means of PeerSim http://peersim.sourceforge.net/), before journal publication. The Reviewer would strongly recommend also open-sourcing the proposed solution to allow researchers and practitioners to run further experiments in either testbed or simulation environments.
  4. Figures – and their description – are still very dense to follow. Probably, fixing point (3) allows also to reduce this section, or to follow it better.
  5. Very few lines for future research work are highlighted in the Conclusions. The Authors should try to add them as bullet points with appropriate descriptions.
  6. A final polishing of English is recommended.

Author Response

Dear Reviewer:

According to your comments, our submitted paper (The editorial reference number for this paper is sensors-1028119 and paper title is Optimal Consensus with Dual Abnormality Mode of Cellular IoT based on Edge Computing) has been revised. However, we hope the revised manuscripts can meet the requirements of publishing on the Sensor.

Best regards to you.

Your Sincerely,

        S.C. Wang,

        scwang@gm.cyut.edu.tw

Point 1: The introduction should be further revised and – possibly – shortened in order to motivate more to the considered problem. A consensus algorithm must guarantee termination, integrity and agreement; however, only integrity is described in the introduction [see https://en.wikipedia.org/wiki/Consensus_(computer_science)]. The Authors should add also termination and agreement, described as separate properties, and relate them to their contribution.

Response 1:

The introduction had been shortened and revised to motivate more to the consensus problem. The constraints of reaching consensus had been modified with three constraints: termination, integrity, and agreement. The properties of termination and agreement had also been added in Abstract, Section 1, 5 and 6, respectively.

Point 2: Background on the considered technologies and network structure would be better separated from Related Work. The Authors should consider creating a “Background” and a “Related Work” sections with the material of Sect. 2. While performing this, the related work section should also be extended to highlight the elements of novelty of the proposed approach with respect to the state of the art, e.g., “compared to [X], OCDAM improves on/differs since/…”.

Response 2:

(1) In Section 2 “Related Works”, the background of consensus problem and the comparisons of consensus protocols in different network topologies have been separated.
(2) The network structure has been separated from the “Related Work” into the Sect. 3. Network Structure.
(3) In Section 2.2., two parts are discussed. One is the comparison of the previous research results of consensus protocols in different network topologies (Table 1); another is to compare the consensus protocols CIoTAP [27] with the proposed OCDAM in ECIoT (Table 2). The symptom of abnormal PE allowed in [27] is malicious only, but OCDAM allows the symptoms of both dormant and malicious to exist in ECIoT simultaneously. Therefore, the fault tolerance capability of our protocol OCDAM is much better than the CIoTAP [27].
The whole descriptions are shown in lines 151-162.

Point 3: With their theoretical contribution, the Authors are addressing problems in a very practical application scenario from 5G networks. For these reasons, to follow a sound scientific methodology (proofs + experiments), the Reviewer strongly believes that their contribution needs to include an implementation of their solution and experimental assessment against meaningful KPI, at least through simulation (e.g. by means of PeerSim http://peersim.sourceforge.net/), before journal publication. The Reviewer would strongly recommend also open-sourcing the proposed solution to allow researchers and practitioners to run further experiments in either testbed or simulation environments.

Response 3:

To focus only on the theoretical study, the practical application of AIoT had been replaced by a pure ECIoT network environment. Underlying the network, an example is taken to simulate the protocol step by step. The seudo code had also been provided in the paper for further simulation or experiments. We hope that the
modification can meet the minimum requirements of
proofs + experement/simulation the referee recommented.
The descriptions are shown in lines 410-424.

Point 4: Figures – and their description – are still very dense to follow. Probably, fixing point (3) allows also to reduce this section, or to follow it better.

Response 4:

All figures in this article have been readjusted for better following.

Point 5: Very few lines for future research work are highlighted in the Conclusions. The Authors should try to add them as bullet points with appropriate descriptions.

Response 5:

(1) In Section 7, the conclusion of our research and the future works had been discussed separated.
(2)The Future Works are added in Section 7.2.
“The fallible component is restricted to abnormal PEs in the paper. It is not enough to realize a high reliable ECIoT. In the more generalized ECIoT, not only PEs may be fallible in the network, but also transmission media may be fallible [30,41,42]. Therefore, in future research, when the abnormal PEs and transmission media are both existed in ECIoT simultaneously, the proposed protocol will be extended to solve the generalized consensus problem.

In addition, in order to maintain the reliability of ECIoT, another related problem called the Fault Diagnosis Protocol (FDA) [43-46] will be discussed in the future. If a protocol can be proposed to help each PE detect and locate abnormal components in ECIoT, then the reliability of ECIoT can be maintained to provide a stable application service environment for CIoT.”

The descriptions are shown in lines 664-672.

Point 6: A final polishing of English is recommended.

Response 6:

The whole paper had been reviewed carefully by a person whose native language is English. The grammar errors are checked again. Some sentences are re-written to improve the readability.

Reviewer 2 Report

What the reviewer points out has been appropriately modified.
However, one round of corrections such as grammatical errors or spell checks are needed.

Author Response

Dear Reviewer:

According to your comments, our submitted paper (The editorial reference number for this paper is sensors-1028119 and paper title is Optimal Consensus with Dual Abnormality Mode of Cellular IoT based on Edge Computing) has been revised. However, we hope the revised manuscripts can meet the requirements of publishing on the Sensor.

Best regards to you.

Your Sincerely,

        S.C. Wang,

        scwang@gm.cyut.edu.tw

Point: However, one round of corrections such as grammatical errors or spell checks are needed.

Response:

The whole paper had been reviewed carefully by a person whose native language is English. The grammar errors are checked again. Some sentences are re-written to improve the readability.